# Robust Principal Component Thermography for Defect Detection in Composites

**DOI:** 10.3390/s21082682

**Published:** 2021-04-10

**Authors:** Samira Ebrahimi, Julien Fleuret, Matthieu Klein, Louis-Daniel Théroux, Marc Georges, Clemente Ibarra-Castanedo, Xavier Maldague

**Affiliations:** 1Computer Vision and Systems Laboratory (CVSL), Department of Electrical and Computer Engineering, Laval University, Quebec City, QC G1V 0A6, Canada; samira.ebrahimi.1@ulaval.ca (S.E.); julien.fleuret.1@ulaval.ca (J.F.); Xavier.Maldague@gel.ulaval.ca (X.M.); 2Infrared Thermography Testing Systems, Visiooimage Inc., Quebec City, QC G1W 1A8, Canada; matthieu.klein@visiooimage.com; 3Centre Technologique et Aérospatial (CTA), Saint-Hubert, QC J3Y 8Y9, Canada; louis-daniel.theroux@cegepmontpetit.ca; 4Centre Spatial de Liège, STAR Research Unit, Liège Université, 4031 Angleur, Belgium; mgeorges@uliege.be

**Keywords:** Robust PCA, RPCA, PCP, OIALM, Orthogonal IALM, noise reduction, pulsed thermography, CFRP

## Abstract

Pulsed Thermography (PT) data are usually affected by noise and as such most of the research effort in the last few years has been directed towards the development of advanced signal processing methods to improve defect detection. Among the numerous techniques that have been proposed, principal component thermography (PCT)—based on principal component analysis (PCA)—is one of the most effective in terms of defect contrast enhancement and data compression. However, it is well-known that PCA can be significantly affected in the presence of corrupted data (e.g., noise and outliers). Robust PCA (RPCA) has been recently proposed as an alternative statistical method that handles noisy data more properly by decomposing the input data into a low-rank matrix and a sparse matrix. We propose to process PT data by RPCA instead of PCA in order to improve defect detectability. The performance of the resulting approach, Robust Principal Component Thermography (RPCT)—based on RPCA, was evaluated with respect to PCT—based on PCA, using a CFRP sample containing artificially produced defects. We compared results quantitatively based on two metrics, Contrast-to-Noise Ratio (CNR), for defect detection capabilities, and the Jaccard similarity coefficient, for defect segmentation potential. CNR results were on average 40% higher for RPCT than for PCT, and the Jaccard index was slightly higher for RPCT (0.7395) than for PCT (0.7010). In terms of computational time, however, PCT was 11.5 times faster than RPCT. Further investigations are needed to assess RPCT performance on a wider range of materials and to optimize computational time.

## 1. Introduction

The unique features that make carbon fiber reinforced plastics (CFRP) preferable to other materials are their high strength-to-weight ratio, good corrosion resistance, high fatigue resistance, and very low coefficient of thermal expansion. These interesting characteristics made CFRP a preferred choice in aerospace and other industries where manufacturing quality, weight, and user safety are of paramount importance. Non-destructive testing (NDT) techniques are regularly used to evaluate the efficiency and locate anomalies non-invasively. infrared thermography (IRT) is a fast, non-contact, and non-invasive NDT approach to detect and characterize anomalies (surface or sub-surface defects) in materials. Pulsed thermography (PT) is one of the most popular active IRT approaches [1]. It is based on thermal heat transfer analysis in transient mode (during cooling).

In PT, a short pulse of energy is applied to the surface of the object being inspected; once the light reaches the sample, it becomes a thermal wave propagating through the material by conduction. An infrared camera records the surface temperature decay. Following the thermal pulse, materials without defects cool down uniformly. However, the presence of a discontinuity will change the diffusion rate, which will affect the heat distribution of the object. These thermal changes will appear at the surface at different times, depending on the properties of the object and the defects, as well as their depths. The deeper is the discontinuity, the later it is observed and the lower is its thermal contrast. However, pulsed thermography data is affected by electronic noise as well as thermal and optical artifacts that may reduce defect detectability. To extract meaningful information from the noisy recorded data, different mathematical methods have been proposed.

Principal component thermography (PCT) [2] is one of such processing methods, which is based on principal component analysis [3]. PCA is a tool for high-dimensional data processing that highlights the similarities and dissimilarities in data and estimates the low-dimensional subspace. As shown in [4], PCA can provide contrast enhancement and flaw depth estimation with a reconstructed data matrix. The goal of this method is to efficiently and accurately estimate the low intrinsic matrix lying on original data. Suppose matrix **M** is a stacked column-vector of data:(1)M=L+S
where L has a low-rank and S is a small perturbation matrix. Classical PCA looks for the rank-*k* estimate of L by solving Equation (Equation 2):(2)$$\begin{array}{c} Minimize∥(M-L)∥\\ subject\;to:\;\;rank(L)⩽K \end{array}$$
where ∥M∥ denotes the 2-norm and K≪min(m,n) is the target dimension of the subspace. In addition, when S is small and independent and identically distributed (i.i.d) Gaussian noise, it is convenient to solve the problem via singular value decomposition (SVD). PCA works efficiently when the data enjoy a low level of noise; otherwise, in highly corrupted data, the estimated **L** can be significantly affected.

Candès et al. [5] proposed an interesting method for rendering PCA more robust, where the objective is to recover the low-rank matrix **L** from highly corrupted measurements **M**. According to Candès et al., unlike classical PCA in which noise should be small, the entries in **S** can have an arbitrarily large magnitude; therefore, it has been used on data containing high levels of noise, outliers, and distortions. To do so, Candès et al. decomposed the input data into a low-rank matrix and a sparse matrix, where the sparse matrix is an estimation of the noise in the data, and the low-rank matrix is an estimation of the data without noise. Candes et al.’s method is known as Robust Principal Component Analysis (RPCA).

RPCA has been used in a wide range of applications such as background estimation and foreground estimation [6], video surveillance [7], face recognition [8], speech recognition [9], latent segmentation indexing [10], and ranking and collaborative filtering to cite a few. Recently, Song et al. [11] employed the RPCA technique for noise reduction and signal enhancement in distributed detection of micro-cracks on structural elements with very small crack opening displacements. The method used by Candès et al. to decompose the data is known as principal component pursuit (PCP). Several other approaches in order to decompose input data into a low-rank matrix and sparse matrix have been proposed [7,12,13,14]. The minimization method used in the PCP has itself been the topic of many studies. Xue et al. [15] and Yang et al. [16] offered two new implementations of the PCP decomposition approach, named Exact Augmented Lagrange Multiplier (EALM) and Inexact Augmented Lagrange Multiplier (IALM). Guyon et al. [14] proposed to solved PCP by using a Linearized Alternating Direction Method with Adaptive Penalty (LADMAP), while Wang et al. studied the accelerated proximal gradient (APG) algorithm to solve it [7].

In this paper, we introduce a new variation for the decomposition by PCP. Then, we compare its performance with respect to the state-of-the-art PT processing techniques.

The paper is organized as follows. Section 2 introduces some recent works regarding both PT data processing as well as NDT applications using RPCA. In Section 3, we present the details of the proposed algorithm. Section 4 details the different aspects of our investigations. Section 5 introduces the results we obtained, while Section 6 analyzes and discusses the results we obtained. Finally, Section 7 concludes this study.

## 2. Literature Review

As previously stated, PT is a field that is eagerly searching for new processing methods to improve the detection of defects. Thus, over the years, several approaches have been proposed. In this section, we briefly introduce methods among the recently proposed.

Khan et al. [17] used a convolutional-auto-encoder-based approach for Intrusion detection, which is fast, simple, and efficient in terms of power and cybersecurity. In addition, Zhang et al. [18] developed an approach combining domain adaption (DA) and adaptive convolutional neural network (ACNN) for steel surface defect detection, which showed an improved accuracy with respect to other methods.

Fleuret et al. [19] investigated the possible application of the Monogenic-Signal to PT data. Promising results were found; nevertheless, the method proved to be highly sensitive to noise. The same year, Yousefi et al. [20] studied an application of Sparse-PCA to PT, under the name SPCT, which outperformed existing methods in terms of defect detection although requiring a significantly higher computational time.

Wen et al. [21,22] used an improved version of the Sparse-PCA to speed up processing. This method named Edge-Group Sparse PCA (ESPCT) [23] was significantly faster than SPCT, although still noticeably slower than PCT. It offered higher defect contrast, making it very promising for the detection of smaller defects in composite materials.

Yousefi et al. [20,24,25] investigated the application of several non-negative matrix factorization (NMF) methods on a wide set of materials. These studies highlighted that NMF offers noticeably better results than other component-based approaches.

The implementation of independent component analysis (ICA) on pulsed thermography inspection of CFRP has been investigated by several authors [26,27,28]. Rengifo et al. [26] reported achieving a sensitivity of 70% on a sample test. Liu et al. [27] highlighted the ability of the ICA to handle thermal inhomogeneities, and other noise sources, as well as providing good defect contrast. Fleuret et al. [28] observed that ICA was comparable to PCT in terms of performance, but with the advantage of being less sensitive to background noise.

Fleuret et al. [29] investigated the application of another approach named Latent Low-Rank Representation (LatLRR) to PT data under the name LatLRRT. This approach differs from most of the previous works because it is based on the assumption that the data are composed of three signals: the observed data, the sparse noise, and the unobserved data. The authors concluded that LatLRRT used as a post-processing method can significantly improve results with respect to state-of-the-art approaches such as PPT [30]. Nevertheless, for the moment, the memory cost prevents this method from being used as a processing method.

Recently, Liu et al. [31] introduced an approach that uses data augmentation generated by the deep-learning models. The assumption was that deep-learning models would be able to learn statistical features from the data. Liu et al.’s method provided good results on composite materials compared with state-of-the-art methods such as PCT [4,32]. The same authors evaluated their work using ICA as a detection method [33]. Similar to the previous one, this method provides good results on composite materials.

Lopez et al. [34,35] proposed partial least square (PLS) regression to improve the general quality of the image sequences. During the regression step, the PLS algorithm can model both spatially and temporally the evolution of the signal. It was originally proposed as a denoising technique allowing synthetic data reconstruction in a manner similar to thermographic signal reconstruction (TSR) [36].

Inspired by these works, Fleuret et al. [37] investigated the use of a pair of support vector machine (SVM) algorithms [38] to enhance defect contrast. The first algorithm computes a regression in the time domain, while the second computes a regression in the space domain. Then, the output sequence is reconstructed from these regressions providing images with enhanced defect contrast.

Even though RPCA is a well-known tool for background-foreground subtraction with improved robustness to noise in several imaging application, it has seldom been investigated in infrared thermography. Zhu et al. [39,40] used RPCA to reduce noise from eddy current pulsed thermography (ECPT) data. Their work was based on the RPCA method proposed by Candès et al. [5]. Liang et al. [41] studied the use of a tensor-based RPCA approach proposed by Lu et al. [42] on ECPT. They concluded that TRPCA is a high accuracy defect extraction algorithm. Xiao et al. [43] studied the application of yet another type of RPCA for data fusion.

The RPCA method we proposed is inspired by the work of Candès et al. [5], however more recent works have been suggested since then. For instance, Peng et al. [44] proposed a highly scalable convex RPCA based on ALM and matrix factorization. Sun et al. [45] proposed a graph-based RPCA. Liu et al. [46] also proposed a graph-based method that has the advantage of being adaptive, ensuring in this way that the local structure of the data is well represented in the low-rank matrix.

Wang et al. [47] introduced the Double RPCA (DRPCA), which offers increased robustness regarding the topology of the regions in the image. In addition, unlike most of the RPCA approaches, which are transductive, DRPCA is an inductive approach, which makes it suitable for online application. Ma et al. [48] offered a review of the most popular RPCA methods used for convex optimization. Van Luong et al. [49] proposed an RPCA method for online application such as background and foreground separation. Cai et al. [50] introduced a rapid RPCA based on an accelerated inexact low-rank estimation.

Several other methods can be found in the literature. We included here a selection of studies on which we based our approach. In particular, our choice regarding the work of Candès et al. [5] is due to the stability and robustness of their approach, which for these reasons has become a reference. Nonetheless, as detailed in next section, our proposed approach is better adapted for the PT applications.

## 3. RPCA via OIALM

Among the different methods proposed in the literature to overcome the limitation of the PCA regarding noisy data, the one proposed by Candès et al. [5] has become very popular. In their work, Candès et al. used a convex optimization; the formulation they used is known as principal component pursuit (PCP). Other formulations and improvements of the PCP, have since been proposed. In particular, the work of Lin et al. [51] has become well-known due to its ability to converge faster than similar methods such as accelerated proximal gradient (APG).

Lin et al. proposed two variations of the PCP formulation using the augmented Lagrangian multiplier (ALM) approach, named Exact Augmented Lagrangian Multiplier (EALM) method and Inexact Augmented Lagrangian Multiplier (IALM). Even though both methods attempt to optimize the sparse and low-rank matrix, their main difference is a condition applied on a set of penalty parameters that allow the IALM algorithm to converge faster than EALM by avoiding the minimization of a sub-problem.

Our approach is inspired by the IALM formulation. Given an observation matrix **D**, which is assumed to be the combination of two matrices, **A** (low-rank) and **E** (a sparse matrix), the straightforward formulation to minimize the energy function is to use the $l_{0}$-norm:(3)$$\begin{array}{c} min\;rank\left(A\right)+\lambda {∥E∥}_{0}\\ D=A+E \end{array}$$
where ${∥\cdot ∥}_{0}$ is the $l_{0}$-norm, that is, the number of the non-zero items of the matrix, and implies the sparsity. λ is the balance parameter to determine the contributions of **A** and **E** in minimizing the objective function. Since Equation (Equation 3) is an NP-hard problem, i.e., at least as hard as the hardest problems in non-deterministic polynomial (NP) time, Candès et al. [5] reformulated this equation into a similar convex optimization problem as follows:(4)$$\begin{array}{c} min_{L,S}\left({∥A∥}_{*}+\lambda {∥E∥}_{1}\right)\\ subject\;to\;\;D=A+E \end{array}$$
where ${∥A∥}_{*}$ and ${∥E∥}_{1}$ are the nuclear norm of **A** and $l_{1}$-norm of **E**, respectively. The balance parameter λ is defined as:(5)$$\lambda =1/\sqrt{max(m,n)}$$
where *m* is the number of rows and *n* is the number of columns of the 2D input matrix. Lin et al. [51] solved Equation (Equation 4) using a generic ALM method, which solves the constrained optimization problem:(6)$$\begin{array}{c} minf(x),\;subject\;to\;\;h(x)=0\\ f:R^{n}\to R,h:f:R^{n}\to R^{m} \end{array}$$

The Lagrange function can be defined as:(7)$$L\left(X,Y,\mu \right)=f\left(X\right)+\langle Y,h\left(X\right)\rangle +\frac{\mu }{2}{∥h\left(X\right)∥}_{F}^{2}$$

According to the Lagrange multiplier method, Equation (Equation 6) can be reformulated to solve the RPCA problem as follows:(8)$$\begin{array}{c} X=\left(A,E\right),f\left(X\right)={∥A∥}_{*}+\lambda {∥E∥}_{1}.\\ h(X)=D-A-E \end{array}$$

The Lagrange function of Equation (Equation 8) is defined as:(9)$$L\left(A,E,Y,\mu \right)={∥A∥}_{*}+\lambda {∥E∥}_{1}+\langle Y,D-A-E\rangle +\frac{\mu }{2}{∥D-A-E∥}_{F}^{2}$$
where Y is the Lagrange multiplier and penalty parameter μ is a positive scalar parameter. The approximate exact augmented Lagrange multiplier algorithm used to solve the RPCA problem is shown in Algorithm 1. $Y_{0}$ has been initialized to $Y_{0}=D/J\left(D\right)$ [52], making the objective function value $\langle Y_{0},D\rangle$ reasonably large. In addition, $J\left(D\right)=max({∥A∥}_{2},\lambda^{-1}{∥Y∥}_{\infty })$, where ${∥.∥}_{\infty }$ is the maximum absolute value of the input matrix. In each iteration after solving $A_{k+1}$ and $E_{k+1}$, the low-rank matrix L is updated by Incremental-PCA [53].

In Step 1 of Algorithm 1, ρ is the learning rate and $\mu_{0}$ is the initialization of the penalty parameter that influences the convergence speed. In [51], it is proven that the objective function of the RPCA problem (Equation (Equation 4)), which is non-smooth, has an excellent convergence property. In addition, it has been proven that, to converge to an optimal solution $\left(A^{*},E^{*}\right)$ of the RPCA problem, it is necessary for $\mu_{k}$ to be non-decreasing and $\sum_{k=1}^{+\infty }\mu_{k}^{-1}=+\infty$. As can be noticed from Equation (Equation 10), our proposed approach differs from the IALM in that the low-rank matrix is projected into an orthogonal space, which is why we named this approach Orthogonal IALM:(10)$$\begin{array}{cc} A_{k+1} & =\underset{u\in R^{n}}{argmax}\;u^{T}A_{k+1}^{T}A_{k+1}u\\ \; & =\underset{u\in R^{n}}{argmax}\;u^{T}Cu \end{array}$$
where u is a unit vector so $u^{T}u=1$. This projects the low-rank value into an orthogonal space while maximizing the variance between the projected data. This step provides relevant information to be projected among the projected matrix’s first dimensions; therefore, it is possible to reduce the computational cost by only keeping γ dimensions, which corresponds to a PCA.
**Algorithm 1: RPCA via the Orthogonal IALM method**
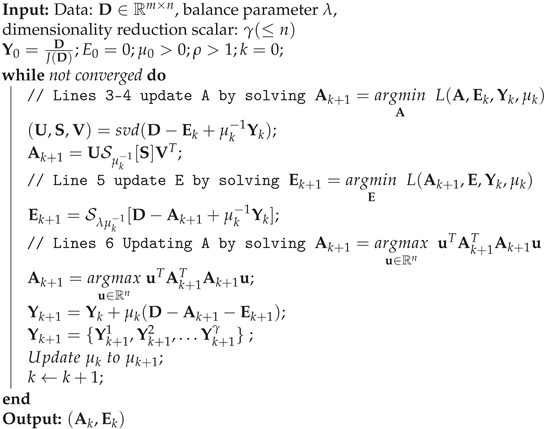


In the next section, we describe our experiments and analysis and present our data.

## 4. Methods

In the previous section, we introduce RPCA. Now, we describe the different aspects of the experiments we conducted in order to evaluate its performance. Figure 1a shows our research block diagram, which presents all of the steps.

### 4.1. Data Acquisition

An academic CFRP plate (30.8 cm × 46 cm × 2.57 mm) was used in this study. It possesses 73 defects of three different types: 23 round flat-bottom holes (FBH), 25 triangular Teflon inserts, and 25 triangular pullouts. These types of manufactured flaws are employed to represent delaminations in CFRP laminates in Ultrasonic Testing because the change in acoustic impedance between the composite and the defect (Teflon or air) produces a variation in the ultrasonic signal when it passes through [54]. This plate was already used to study the adequateness of such artificial defects to represent delamination in thermography and shearography NDT [55] as well as for research on PCA analysis in shearography [56,57]. Defect size, depth, and thickness are indicated in Table 1, and their respective locations are shown in Figure 2a.

The PT experimental setup consists of two flash lamps (5 ms thermal pulse, 6.4 KJ/flash (Balcar, France)), a cooled infrared camera (FLIR Phoenix (FLIR Systems, Inc., Wilsonville, OR, USA), InSb, midwave, 3–5 mm, Stirling Cooling), and a computer to store the thermal sequences. The data were acquired at a frequency of 180 Hz. A control unit was also required to control and synchronize the data acquisition with flash triggering. Our experiments were performed using a PC (Intel(R) Xeon(R), 128 Gb memory, (Intel Corporation, Santa Clara, CA, USA)). Figure 2b, shows the PT approach in reflection mode.

### 4.2. Metrics

In this section, we briefly introduce the different metrics we used to quantitatively assess the performance of our proposed approach RPCT, and compare it to PCT.

#### 4.2.1. Contrast to Noise Ratio (CNR)

The Signal-to-noise ratio (SNR) is a metric that measures image quality by estimating the signal level with respect to the background noise. The Contrast-to-noise ratio (CNR) is similar to SNR although based on the difference (i.e., the contrast) between two features in an image. This contrast can be calculated, for instance, for a defect area with respect to a sound area. This is interesting since it provides a tool to quantitatively assess the defect detection capabilities of a given method. Several CNR definitions can be found in the literature, as summarized by Usamentiaga et al. [58]. This study also proposes to use the following definition, as it has been shown to be the most robust against noise and image enhancement operations (e.g., Gamma correction):(11)$$CNR=\frac{|\mu_{S}-\mu_{N}|}{\sqrt{\frac{\left(\sigma_{S}^{2}+\sigma_{N}^{2}\right)}{2}}}$$
where $\mu_{S}$ is average level of the signal in the defect region, $\mu_{N}$ is the average level of the noise in the sound area, $\sigma_{S}$ is standard deviation of the signal in the defect area, and $\sigma_{N}$ is standard deviation of the noise in the reference region.

#### 4.2.2. Jaccard Similarity Coefficient Score

The Jaccard similarity coefficient (also known as Jaccard index or Intersection-Over-Union (IoU)), initially proposed by Paul Jaccard [59], is a statistical method that measures the similarity between two datasets and is defined as the ratio of the intersection size to the union size of two sample sets (as illustrated in Figure 3).The Jaccard index is a useful and very straightforward metric.

In this approach, four steps should be taken into account:Count the number of members (i.e., pixels) that are shared between both sets (intersection).Count the total number of members in both sets i.e., the union (shared and unshared).Divide the number of shared members (1) by the total number of members (2).Multiply the computed result from Step 3 by 100.

J(A,B) provides a value between 0 (no similarity) and 1 (identical sets). Hence, the higher is the value of IoU, the higher is the level of similarities between two sets (Figure 3b).
(12)$$\begin{array}{c} J\left(A,B\right)=\frac{\left|A\cap B\right|}{\left|A\cup B\right|}=\frac{\left|A\cap B\right|}{\left|A|+|B|-|A\cap B\right|}\;\\ 0\leq J(A,B)\leq 1 \end{array}$$

### 4.3. Analysis

We chose to compare our approach RPCT that is based on RPCA, with principal component thermography (PCT) [32] that is based on PCA, to verify if the use of RPCA will effectively reduce the impact of noise.

The calculation of the score for each one of our two metrics was carried out using two different protocols. For the calculation of the Jaccard Index, we developed a custom automatic segmentation approach, as illustrated in Figure 4. This approach is based on four steps: First, the image’s contrast is corrected using a percentile normalization between the second and ninety-eighth percentiles. The choice of the percentile normalization is due to the ability of this approach to increase the contrast even in the presence of local light artifacts. Then, a bilateral filter [60] is used to smooth the image. To smooth efficiently, we selected a spatial filter with a kernel of size 31×31 pixels and a range filter with kernel of size 7×7 pixels. The third step consists in applying a local thresholding approach based on a block of 101×101 pixels. The last step consists in removing the small objects in 2 connectivity with smaller than 64 pixels present in the image. The reason for the latter step is to remove artifacts that could have been generated by the third step.

Once the images of the different methods are segmented, they can be compared with a manually labeled reference image in order to compute the metric score.

Regarding the CNR score, before the experiments, both the defect regions and sound areas were manually labeled, as can be seen in the two examples shown in Figure 5, where three boundaries can be seen: red, green and blue. The sound area is located in between the outermost boundary (in red) and the middle boundary (in green), while the defect area corresponds to the area inside the innermost boundary (in blue). The average and standard deviation values required in Equation (Equation 11) are calculated from these areas. The exact same areas were used to estimate the CNR values from raw, PCT, and RPCT data.

The following section presents the results.

## 5. Results

The original PT sequence (raw data) was processed by PCT and RPCT. Figure 6 shows some representative results (selected arbitrarily) of the different methods.

Raw data correspond to the original unprocessed data. The arithmetical difference between a defective and a non-defective (sound) pixel (or group of pixels) is called the absolute thermal contrast, or simply the contrast. A given defect would be visible, i.e., its contrast would be higher than zero during a certain time that depends on different factors (defect type, size and depth, inspection method, amount of delivered energy, etc.). The thermal contrast typically varies from zero to a maximum, and then it gradually returns to zero. For instance, considering defect FBH-4M (at the center of the plate, highlighted in Figure 7), Figure 7 shows the thermal profiles, i.e., temperature vs. time plots, for a pixel inside the defect area (red plot), the profile of a pixel in the sound pixel close to this defect (blue plot), and the thermal contrast (green plot).

Hence, defects are visible at different degrees during several frames. In the case illustrated in Figure 7, defect FBH-4M can be detected roughly from 0.1 to 10 s, which corresponds to the time range where the contrast is higher than the noise. All other defects in the inspected plate produce similar profiles, with deeper defects appearing later and with lesser contrast.

PCT and RPCT, on the other hand, are statistical methods based on PCA and RPCA, respectively, that reorder data according to their variability and project them into an orthonormal space in such a way that the most valuable information is compressed in the first few components while subsequent components contain mostly noise. Figure 8 shows some selected PCT components: (Figure 8a) first PCT component showing several defects but also a strong non-uniform heating pattern (two ellipses, left and right) that accounts for most of the variability of the data and are therefore concentrated in the first component; (Figure 8b,c) second and third components, respectively, showing most of the defects and lesser impact from non-uniform heating; and (Figure 8d,f) later components (21st, 500th, and 3500th, respectively) showing mostly noise.

Table 2 summarizes the computational time for each of the processing techniques. As can be seen, RPCT takes significantly more time to compute (483 s) than PCT (42 s). Further optimization of the processing algorithm would be required to reduce this gap.

The CNR values of all defects and all processing techniques were calculated using the defects and reference areas as the ones shown in Figure 5. The highest CNR value (CNRmax) was then found for each case and the results are summarized in Table 3. The maximum CNR values between different methods are in bold.

Figure 9 presents selected CNR curves for raw, PCT, and RPCT data that correspond to defects at the same depth (Z = 1.43 mm) but of different types (FBH, pull-outs, and inserts) in which the CNRmax value is indicated together with the frame of occurrence.

To further analyze results, the CNRmax values are gathered by defect type in Figure 10 and by defect depth in Figure 11.

Figure 10 shows the evolution of the CNR as a function of the depth and each processing method for each defect type. Similarly, Figure 11 shows the evolution of the CNR as a function of the material type for the raw data and each processing method for some selected defects.

Finally, Table 4 presents the best Jaccard Index score obtained for each processing method computed through time.

In the next section, the results are analyzed and discussed.

## 6. Discussion

As can be seen in Table 2, the proposed RPCT method has, for this specific and given dataset, a processing time that is considerably longer than PCT (11.5 times). This can be explained by the convex optimization operation. More precisely, to compute the low-rank and sparse matrix, several SVD are computed at each iteration of the optimization loop until it converges. Nonetheless, despite the increased time compared with the state-of-the-art methods, a computation time of 8 min 3 s is still reasonable for most NDT applications.

Approaches such as General-purpose computing on graphics processing units (GPGPU) can significantly reduce the computation time; however, such implementation exceeded this study’s goals.

Several observations can be made from the results in Figure 10 and Figure 11:In general, flat-bottom-holes (FBHs) present by far the highest CNRmax values, as expected, while pull-outs (POs) and Teflon inserts (TEFs) are very close, although with slightly higher values for the latter contrary to what was expected. Delamination-like artificial defects such as pull-outs should, in principle, present a higher thermal contrast than Teflon inserts, given that the thermo-physical properties of Teflon are closer to those of CFRP than those of air. It can be concluded that these two types of artificial defects are not different enough to produce a noticeable variation in CNR.In the case of FBHs at the same depth, larger defects have slightly higher CNR values than smaller defects, i.e., FBHs with *D* = 12.7 showed higher CNRmax than FBHs with *D* = 6.35 mm (as expected).For pullouts at the same depth, thicker defects have higher CNR values than thinner defects, i.e., *th* = 0.15 vs. 0.10 mm (as expected).Regarding the relative depths, in all cases (FBHs, POs, and TEFs), the deeper is the defect, the lower is the CNR value (as expected).The improvement in CNRmax score after processing (RPCT and PCT) is generally more pronounced for deeper depths.CNR values (considering all defect types) were on average 40% higher for RPCT compared to PCT, which may be taken as an indication of global performance improvement thanks to the use of RPCA.In the case of FBHs, CNRmax values were 60% higher for RPCT vs. PCT.In the case of POs, CNRmax values were 27% higher for RPCT vs. PCT.In the case of TEFs, CNRmax values were 24% higher for RPCT vs. PCT.

In the next section, we provide our conclusion regarding the proposed approach.

## 7. Conclusions

In this paper, we propose a new formulation of the RPCA named Orthogonal Inexact Augmented Lagrangian Multiplier (OIALM). We evaluated the performance of the resulting approach, Robust Principal Component Thermography (RPCT), for detecting defects and discontinuities in CFRP and compared the results with those of PCT using two quantitative metrics: Contrast-to-Noise Ratio (CNR) and the Jaccard similarity coefficient. The CNR was computed frame by frame and the maximum value was identified for every defect and all techniques (raw, PCT, and RPCT).

The low-rank and sparse matrices in RPCT are projections of thermal data onto the noiseless data and noise, and the low-rank matrix is optimized in each iteration using incremental PCA. RPCT enables noise to be removed from the thermal sequences, therefore improving defect contrast and increasing defect detection.

Considering CNR results from all defect types, RPCT reported CNRmax values 40% higher than PCT. The improvement was greater for flat-bottom-holes (60%) compared to pull-outs (27%) and Teflon inserts (24%). In the case of the Jaccard index, RPCT performed slightly better than PCT (0.74 vs. 0.70). The computation time was however 11.5 times longer for RPCT than for PCT.

Although RPCT, to the best of our knowledge, has never been applied to pulsed thermographic data before, this study demonstrates its efficiency for defect enhancement capabilities over mixed and various types of defects typically addressed in IRT in composite materials. The goal of the study was to introduce this approach; further improvement in terms of computational speed could be achieved by using low-level programming language and hardware optimization. RPCT can therefore be considered as a powerful analysis tool that may help to push the limit of defect detection by IRT.

It should be pointed out that the RPCT method is not limited to data acquired by pulsed thermography (in which only the cooling phase is analyzed), but it could potentially be applied to other IRT techniques such as square pulse thermography (in which the heating and cooling phase may be of interest) and lockin thermography (in which regular periodic cycles are observed).

Future research will be directed towards the application of RPCT to different materials (glass fibers, aluminium, etc.), to the implementation of software/hardware optimization solutions (e.g., through the use of GPGPUs), and to the application of RPCT to other IRT techniques (e.g., square pulse thermography and lockin thermography).

## Figures and Tables

**Figure 1 sensors-21-02682-f001:**
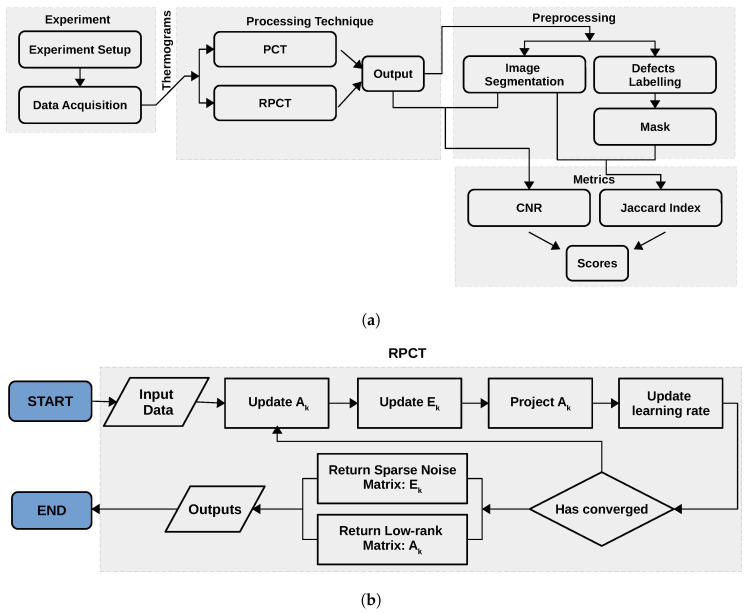
(**a**) Research block diagram; and (**b**) proposed method block diagram.

**Figure 2 sensors-21-02682-f002:**
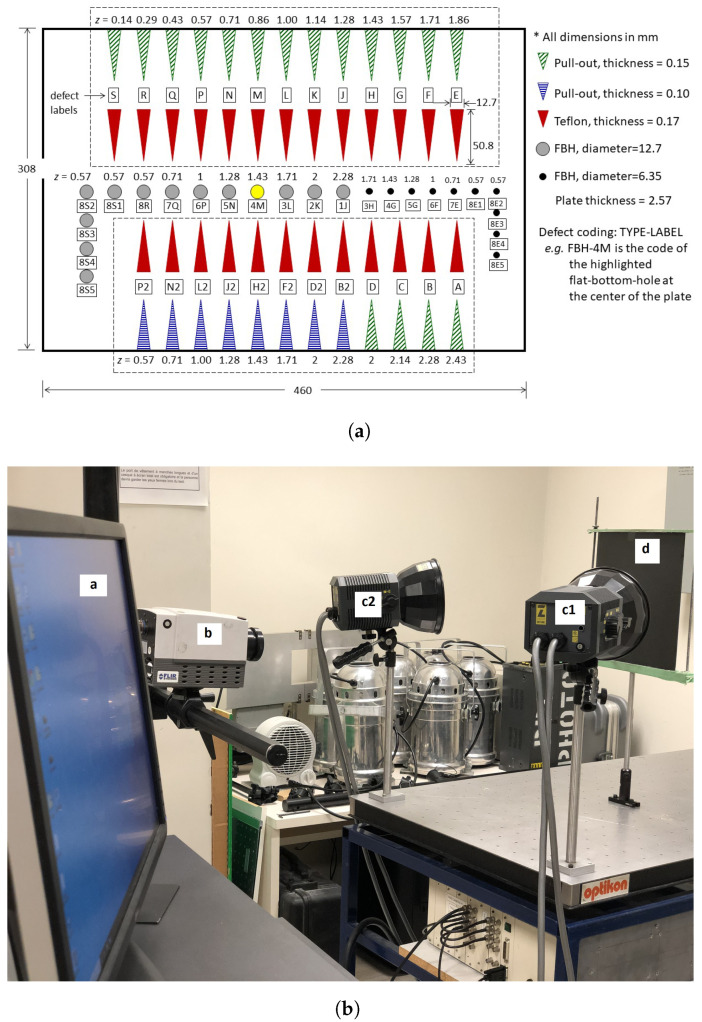
(**a**) CTA CFRP plate, where Z is the defect depth and labels are used to identify the location of each defect; and (**b**) pulsed thermography setup. a, PC; b, IR camera; c1 and c2, left and right flashes; d, CFRP specimen.

**Figure 3 sensors-21-02682-f003:**
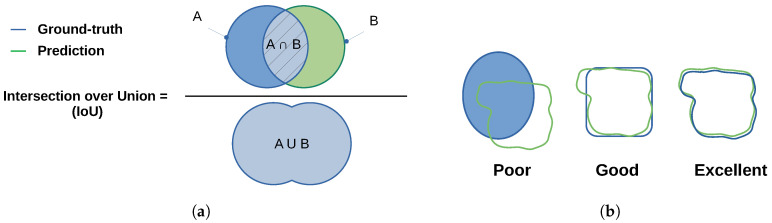
(**a**) Jaccard Index similarity definition; and (**b**) similarity between the ground-truth and the detected area.

**Figure 4 sensors-21-02682-f004:**
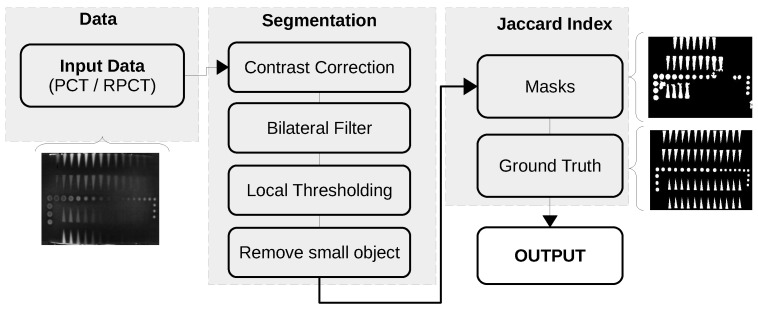
Segmentation and Jaccard index computation flow graph.

**Figure 5 sensors-21-02682-f005:**
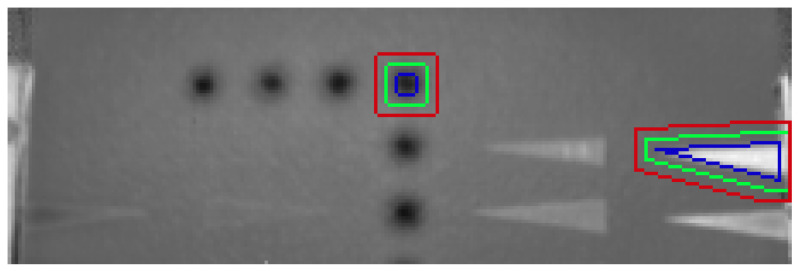
Examples of reference and defect regions. The boundaries of the reference region are between the green and red lines, while the defective region is inside the blue line area.

**Figure 6 sensors-21-02682-f006:**
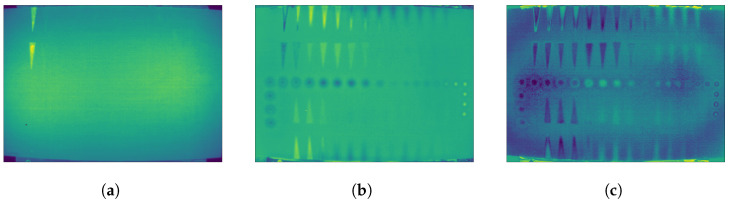
(**a**) Raw data at Frame 5; (**b**) fifth component of PCT data; and (**c**) fifth component of RPCT data.

**Figure 7 sensors-21-02682-f007:**
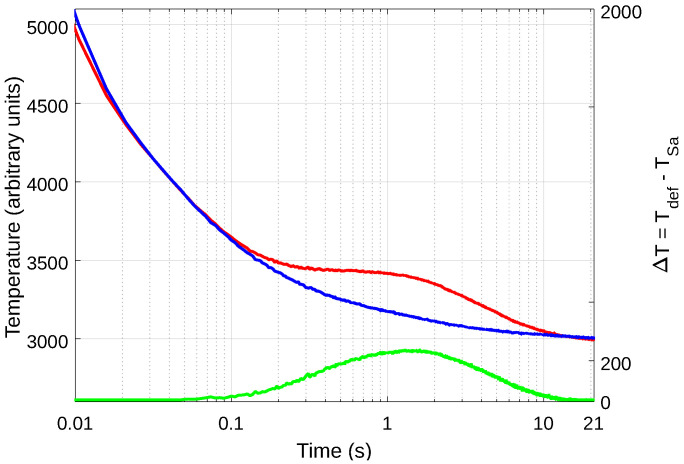
Thermal profiles (temperature vs. time plots on a semi-logarithmic scale) of a pixel inside defect FBH-4M (red plot); a sound pixel close to this defect (blue plot); and the absolute thermal contrast between these two (green plot).

**Figure 8 sensors-21-02682-f008:**
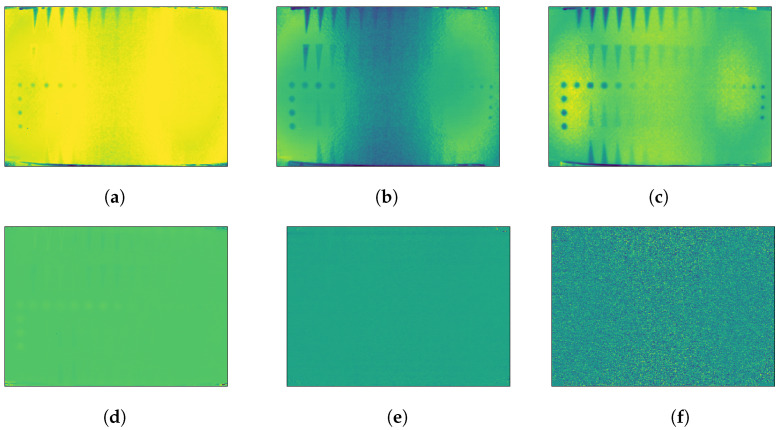
PCT components: (**a**) first; (**b**) second; (**c**) third; (**d**) 21st; (**e**) 500th; and (**f**) 3500th.

**Figure 9 sensors-21-02682-f009:**
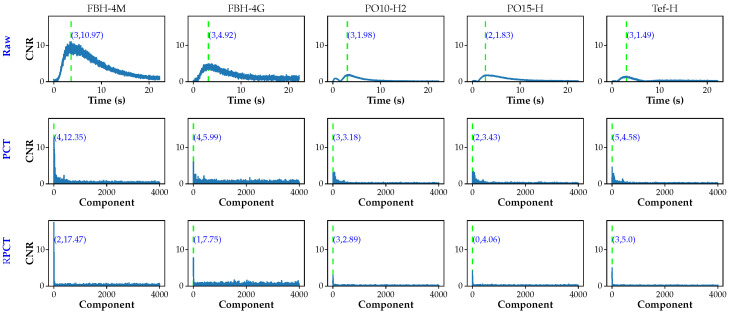
Comparative CNR curves for defects at the same depth (Z = 1.43 mm) but different types (FBH, pull-outs and inserts): (**Row 1**) raw data; (**Row 2**) PCT; and (**Row 3**) RPCT. Columns 1–5 represent FBH-4M, FBH-4G, PO10-F2, PO15-H, and Tef-H, respectively.

**Figure 10 sensors-21-02682-f010:**
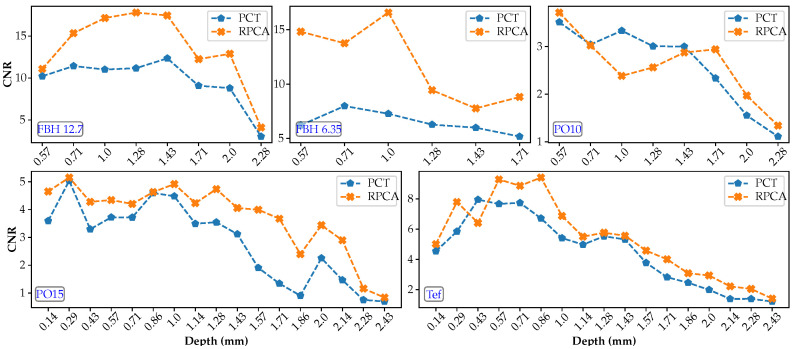
CNRmax by defect type as a function of the defect depth for all data sequences (PCT and RPCT).

**Figure 11 sensors-21-02682-f011:**
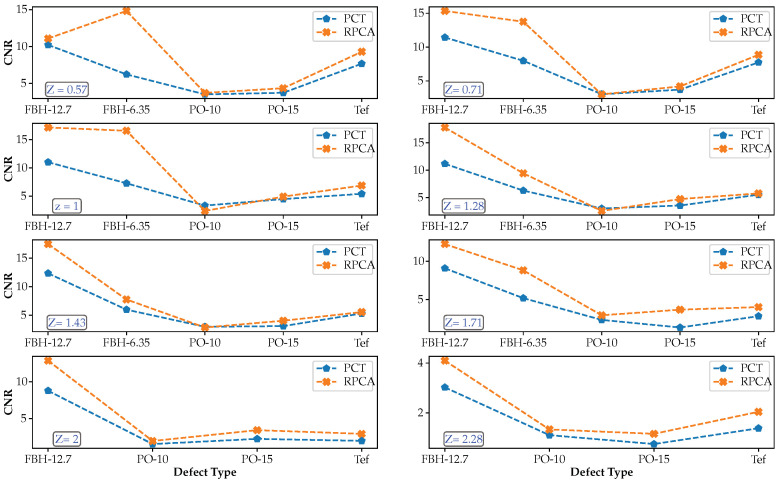
CNRmax by defect depth as a function of the defect type for all data sequences (PCT and RPCT).

**Table 1 sensors-21-02682-t001:** Defect specifications for the CFRP Plate, Z is the depth of the defect below the inspected surface. Thickness is the defect thickness or thickness of the holes in case of the FBH type of defect.

Defect Code	Z [mm]	Dimensions [mm]	Thickness [mm]	Defect Code	Z [mm]	Dimensions [mm]	Thickness [mm]	Defect Code	Z [mm]	Dimensions [mm]	Thickness [mm]
**Teflon Inserts**				**Pull-Outs**				**FlatBottom Holes**			
Tef-A	2.43	12.7 × 50.8	0.17	PO15-A	2.43	12.7 × 50.8	0.15	FBH-1J	2.28	12.70	0.29
Tef-B	2.28	12.7 × 50.8	0.17	PO15-B	2.28	12.7 × 50.8	0.15	FBH-2K	2.00	12.70	0.57
Tef-C	2.14	12.7 × 50.8	0.17	PO15-C	2.14	12.7 × 50.8	0.15	FBH-3L	1.71	12.70	0.86
Tef-D	2.00	12.7 × 50.8	0.17	PO15-D	2.00	12.7 × 50.8	0.15	FBH-4M	1.43	12.70	1.14
Tef-E	1.86	12.7 × 50.8	0.17	PO15-E	1.86	12.7 × 50.8	0.15	FBH-5N	1.28	12.70	1.29
Tef-F	1.71	12.7 × 50.8	0.17	PO15-F	1.71	12.7 × 50.8	0.15	FBH-6P	1.00	12.70	1.57
Tef-G	1.57	12.7 × 50.8	0.17	PO15-G	1.57	12.7 × 50.8	0.15	FBH-7Q	0.71	12.70	1.86
Tef-H	1.43	12.7 × 50.8	0.17	PO15-H	1.43	12.7 × 50.8	0.15	FBH-8R	0.57	12.70	2.00
Tef-J	1.28	12.7 × 50.8	0.17	PO15-J	1.28	12.7 × 50.8	0.15	FBH-8S1	0.57	12.70	2.00
Tef-K	1.14	12.7 × 50.8	0.17	PO15-K	1.14	12.7 × 50.8	0.15	FBH-8S2	0.57	12.70	2.00
Tef-L	1.00	12.7 × 50.8	0.17	PO15-L	1.00	12.7 × 50.8	0.15	FBH-8S3	0.57	12.70	2.00
Tef-M	0.86	12.7 × 50.8	0.17	PO15-M	0.86	12.7 × 50.8	0.15	FBH-8S4	0.57	12.70	2.00
Tef-N	0.71	12.7 × 50.8	0.17	PO15-N	0.71	12.7 × 50.8	0.15	FBH-8S5	0.57	12.70	2.00
Tef-P	0.57	12.7 × 50.8	0.17	PO15-P	0.57	12.7 × 50.8	0.15	FBH-3H	1.71	6.35	0.86
Tef-Q	0.43	12.7 × 50.8	0.17	PO15-Q	0.43	12.7 × 50.8	0.15	FBH-4G	1.43	6.35	1.14
Tef-R	0.29	12.7 × 50.8	0.17	PO15-R	0.29	12.7 × 50.8	0.15	FBH-5G	1.28	6.35	1.29
Tef-S	0.14	12.7 × 50.8	0.17	PO15-S	0.14	12.7 × 50.8	0.15	FBH-6F	1.00	6.35	1.57
Tef-B2	2.28	12.7 × 50.8	0.17	PO10-B2	2.28	12.7 × 50.8	0.10	FBH-7E	0.71	6.35	1.86
Tef-D2	2.00	12.7 × 50.8	0.17	PO10-D2	2.00	12.7 × 50.8	0.10	FBH-8E1	0.57	6.35	2.00
Tef-F2	1.71	12.7 × 50.8	0.17	PO10-F2	1.71	12.7 × 50.8	0.10	FBH-8E2	0.57	6.35	2.00
Tef-H2	1.43	12.7 × 50.8	0.17	PO10-H2	1.43	12.7 × 50.8	0.10	FBH-8E3	0.57	6.35	2.00
Tef-J2	1.28	12.7 × 50.8	0.17	PO10-J2	1.28	12.7 × 50.8	0.10	FBH-8E4	0.57	6.35	2.00
Tef-L2	1.00	12.7 × 50.8	0.17	PO10-L2	1.00	12.7 × 50.8	0.10	FBH-8E5	0.57	6.35	2.00
Tef-N2	0.71	12.7 × 50.8	0.17	PO10-N2	0.71	12.7 × 50.8	0.10				
Tef-P2	0.57	12.7 × 50.8	0.17	PO10-P2	0.57	12.7 × 50.8	0.10				

**Table 2 sensors-21-02682-t002:** Processing time of RPCT and PCT methods.

Process	RPCT	PCT
Run time (s)	483.06	42.007
ROI size (px)	229 × 320	229 × 320
Number of frames	3998	3998

**Table 3 sensors-21-02682-t003:** Maximum CNR (CNRmax) values for RPCT and PCT methods in all depths (Z).

Teflon		CNRmax			FBH D = 12.7 mm		CNRmax			PO th = 0.15 mm		CNRmax		
Code	Z	RPCT	PCT	RPCT vs. PCT	Code	Z	RPCT	PCT	RPCT vs. PCT	Code	Z	RPCT	PCT	RPCT vs. PCT
Tef-A	2.43	**1.408**	1.214	16%	FBH-1J	2.28	**4.098**	3.025	35%	PO15-A	2.43	0.839	0.697	20%
Tef-B	2.28	**2.0465**	1.3865	48%	FBH-2K	2	**12.887**	8.795	47%	PO15-B	2.28	**1.163**	0.754	54%
Tef-B2	2.28				FBH-3L	1.71	**12.255**	9.081	35%	PO15-C	2.14	**2.897**	1.466	98%
Tef-C	2.14	**2.216**	1.385	60%	FBH-4M	1.43	**17.464**	12.346	41%	PO15-D	2	**3.439**	2.254	53%
Tef-D	2	**2.937**	1.988	48%	FBH-5N	1.28	**17.808**	11.174	59%	PO15-E	1.86	**2.399**	0.908	164%
Tef-D2	2				FBH-6P	1	**17.155**	11.012	56%	PO15-F	1.71	**3.673**	1.343	173%
Tef-E	1.86	**3.085**	2.457	26%	FBH-7Q	0.71	15.347	11.432	34%	PO15-G	1.57	**3.994**	1.906	110%
Tef-F	1.71	**4.008**	2.8225	42%	FBH-8R	0.57	11.084	10.2143	9%	PO15-H	1.43	4.057	3.118	30%
Tef-F2	1.71				FBH-8S1	0.57				PO15-J	1.28	**4.734**	3.545	34%
Tef-G	1.57	4.581	3.778	21%	FBH-8S2	0.57				PO15-K	1.14	**4.231**	3.493	21%
Tef-H	1.43	**5.5655**	5.3225	5%	FBH-8S3	0.57				PO15-L	1	4.916	4.479	10%
Tef-H2	1.43				FBH-8S4	0.57				PO15-M	0.86	4.63	4.597	1%
Tef-J	1.28	**5.7675**	5.531	4%	FBH-8S5	0.57				PO15-N	0.71	**4.202**	3.718	13%
Tef-J2	1.28				**FBH D = 6.35 mm**		**CNRmax**			PO15-P	0.57	**4.344**	3.722	17%
Tef-K	1.14	5.507	4.979	11%	**Code**	**Z**	**RPCT**	**PCT**	**RPCT vs. PCT**	PO15-Q	0.43	**4.278**	3.295	30%
Tef-L	1	**6.8795**	5.412	27%	FBH-3H	1.71	8.817	5.182	70%	PO15-R	0.29	**5.151**	5.035	2%
Tef-L2	1				FBH-4G	1.43	**7.769**	5.993	30%	PO15-S	0.14	**4.65**	3.595	29%
Tef-M	0.86	**9.418**	6.719	40%	FBH-5G	1.28	**9.44**	6.277	50%	**PO th = 0.15 mm**		**CNRmax**		
Tef-N	0.71	**8.8715**	7.744	15%	FBH-6F	1	**16.581**	7.276	128%	**Code**	**Z**	**RPCT**	**PCT**	**RPCT vs. PCT**
Tef-N2	0.71				FBH-7E	0.71	**13.755**	7.985	72%	PO10-B2	2.28	**1.34**	1.111	21%
Tef-P	0.57	**9.292**	7.6725	21%	FBH-8E1	0.57	**14.8234**	6.2194	138%	PO10-D2	2	**1.969**	1.552	27%
Tef-P2	0.57				FBH-8E2	0.57				PO10-F2	1.71	**2.939**	2.334	26%
Tef-Q	0.43	6.417	**7.953**	−19%	FBH-8E3	0.57				PO10-H2	1.43	2.873	**2.998**	−4%
Tef-R	0.29	7.799	5.851	33%	FBH-8E4	0.57				PO10-J2	1.28	2.561	**3.005**	−15%
Tef-S	0.14	5.016	4.542	10%	FBH-8E5	0.57				PO10-L2	1	2.383	**3.33**	−28%
										PO10-N2	0.71	3.021	**3.041**	−1%
										PO10-P2	0.57	**3.711**	3.51	6%

**Table 4 sensors-21-02682-t004:** Jaccard Index values of RPCT and PCT segmentation.

Method	RPCT	PCT
**Jaccard Index**	0.7395	0.7010

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
