# Peer review of "Robust Principal Component Thermography for Defect Detection in Composites"

_sensors, 2021, doi:10.3390/s21082682_

Round 1

Reviewer 1 Report

Really good paper, congrats to the authors.

I have only some minor things to be checked and corrected:

Figure 5 – should be „3rd image “, not „3th image”

Page 11 –  it should be “concluded then that these two” not “concluded then than these two”

Page 11 –  it should be “it must be assessed on a wider range” not “it must be assess on a wider range”

Reference 24 – please complete the reference, it's missing some information like publisher etc.

Author Response

Dear Reviewer,

You will find attached our reply to your report.

Thank you very much for your time and evaluation.

Kind Regards,

Clement Ibarra-Castanedo and Co-authors

Reviewer 2 Report

The study presents signal processing method to enhance NDE of CFRP composites with the pulsed tomography. This research is of significance, however, I find that the proposed method is not strong enough to justify it's choice over the other methods that it was compared witch. The manuscript has to be revised and can be resubmitted. Authors are encouraged to answer the following comments.

1) Equation (5): What do m and n mean? There is no description of these variables.

2) Algorithm 1: What are mu_0 and rho values and what purpose do they serve?

3) Subsection 3.1: A reference must be put after stating that those defects block ultrasound propagation.

4) Figure 1a: This figure is a little confusing. What do the letters and numbers, for example, 2K, 1J, 3H, ... mean? Why is FBH hole with larger diameter shown as 0.10 and the one with smaller diameter as 6.35?

5) Equation (11): How are mu_N and sigma_N calculated? This is not mentioned in any way. Calculation of mu_S implies averaging signal levels over a region of defect. But how is the region of defect defied? Does this region have sharp, well-defined boundaries? Or are boundaries smeared? I see a partial answer in Figure 4, but how was the size of a reference region selected?

6) Figure 6: Why has maximum CNR occurred at different times for different methods? Shouldn't the time of max CNR for a raw image be set as a reference for other methods and the CNR be calculated for them at that particular time instant? In that case, the max CNR values for all the methods would be lower.

7) Table 4: I reccommend to test the statistical significance of differences of the Jaccard index among the methods. These values are quite close to each other. IT may be that the difference between them is not really significant.

8) What are the merits of RPCT compared to PPT and PCT? It showed a little higher index scores, but the computational time is much higher and the method itself seems quite complex. Authors mentioned that it can be improved with hardware and programming, but is it also not the case for those other two methods? In the end, I am not convinced that the RPCT is worth using instead of more established PPT and PCT.

Author Response

(The authors gave the same response as above.)

Reviewer 3 Report

In this study authors used; Robust Principal Component Analysis (RPCA) is investigated as a means to process Pulsed Thermography (PT) data. We limited our study to the Carbon-Fiber Reinforced Plastic (CFRP) material due to its wide range of applications. We found that existing state-of the- art RPCA methods could be improved for our application, and therefore we introduced a new variation based on the Principal Component Pursuit (PCP) framework. We compared our approach, RPC thermography (RPCT), with advanced PT data processing methods based on two metrics, Contrast-to-Noise Ratio (CNR) and the Jaccard similarity coefficient. The paper is interesting overall, but the following are the comments that must be addressed:

 Comments:

English should be corrected.

  • please add a block diagram of the proposed research step by step, what is the result of the paper?

please add a block diagram of the proposed method.

please add sentences about future analysis.

Figures should have better quality; for now, they are low.

Fonts of the figures should be bigger.

Formulas and fonts should be formatted.

References should be 2018-2021 Web of Science about 50% or more ;30 at least.

Please compare with other methods, justify. Advantages or Disadvantages different methods

  • Authors need to re-write the Abstract in a more meaningful way example (Problem definition=> How existing methods are lacking => proposed solution => Outcome). ADD the accuracy of the proposed approach in number in Abstract.
  • Authors need to elaborate, why Robust Principal Component Analysis (RPCA) selected, (Technical reasons must be provided.
  • Section 1 introduction part is the most important part of the manuscript and the authors did not pay any attention to it. it needs to improve authors summed up with outdated References which are not efficient.

Song, Qingsong, Guoping Yan, Guangwu Tang, and Farhad Ansari. "Robust principal component analysis and support vector machine for detection of microcracks with distributed optical fiber sensors." Mechanical Systems and Signal Processing 146 (2021): 107019. Liu, Kaixin, Yuwei Tang, Weiyao Lou, Yi Liu, Jianguo Yang, and Yuan Yao. "A thermographic data augmentation and signal separation method for defect detection." Measurement Science and Technology 32, no. 4 (2021): 045401. Khan, M.A. and Kim, J., 2020. Toward Developing Efficient Conv-AE-Based Intrusion Detection System Using Heterogeneous Dataset. Electronics, 9(11), p.1771.Zhang, Siyu, Qiuju Zhang, Jiefei Gu, Lei Su, Ke Li, and Michael Pecht. "Visual inspection of steel surface defects based on domain adaptation and adaptive convolutional neural network." Mechanical Systems and Signal Processing 153 (2021): 107541.

  • The major contribution of the article is not clear.
  • Related work section is missing??
  • Before Conclusion, please draw a Table and compare it with previous researchers, How your approach is better in terms of accuracy.
  • Authors miss experiment setup, without experimental setup results are doubtful please explain the experimental environment in detail.
  • Conclusion: point out what are you done.

-is there a possibility to use the proposed method for other problems?

Author Response

(The authors gave the same response as above.)

Round 2

Reviewer 2 Report

I would like to express my gratitude to the authors for improving the manuscript. Now the manuscript is worthy of publication.

One small remark, though. The authors commented that the region of defect and surrounding region in Figure 5 (in revised version) were selected manually. Why were these regions selected the way they were? What was the scientific basis for selecting these particular pixel values to define the defect boundaries?

Author Response

We deeply appreciate your comments and suggestions, which we believe allowed us to significantly improve our work.

We selected these regions manually for simplicity. We tried automatic detection of defects in a previous work (please, see reference [58]) using a simpler specimen. In the present case, automatic segmentation was out of the scope (and also more complicated to apply on a specimen having 73 defects with different shapes, some of which at the border of the plate), the purpose was to compare two methods (RPCT vs. PCT) so we opted to define manually the defective and sound areas (which we knew of course).

There are no standards for the calculation of the CNR specifically for infrared thermography. There exists however, some norms related to other NDT techniques, for example radiology (ASTM E2737), from which we got inspired for the definition of defect and sound areas. There are two main things: (1) each defect has its own sound area neighboring it to reduce the impact of artifacts (such as non-uniform heating in the case of IRT); and (2) it is assumed that the defect is surrounded by a “transition area” (between the defect and the sound areas) that may be affected by the proximity of the defects, so it should be avoided. The standard indicates that the sound area should be twice the size of the defect area. In our case however, this was not possible to achieve given the proximity of defects. Hence, we selected areas as large as possible.

We sincerely hope we properly answered your questions.

Reviewer 3 Report

The authors did excellent work and resolve almost all my previous queries now this paper looks very good and interesting for readers so I agree to accept it in its present form.

Author Response

We wish to thank the Reviewer for the valuable inputs that indeed allowed us to produce an improved paper.